# Influence of the Charge Ratio of Guanine-Quadruplex Structure-Based CpG Oligodeoxynucleotides and Cationic DOTAP Liposomes on Cytokine Induction Profiles

**DOI:** 10.3390/biom13111639

**Published:** 2023-11-11

**Authors:** Nguyen Bui Thao Le, Anh Thi Tram Tu, Dandan Zhao, Chiaki Yoshikawa, Kohsaku Kawakami, Yoshihisa Kaizuka, Tomohiko Yamazaki

**Affiliations:** 1Research Center for Macromolecules and Biomaterials, National Institute for Materials Science (NIMS), 1-2-1 Sengen, Tsukuba 305-0047, Japan; le.buithaonguyen@nims.go.jp (N.B.T.L.); tuthitramanh@gmail.com (A.T.T.T.); amberdiary@gmail.com (D.Z.); yoshikawa.chiaki@nims.go.jp (C.Y.); kawakami.kohsaku@nims.go.jp (K.K.); kaizuka.yoshihisa@nims.go.jp (Y.K.); 2Division of Life Science, Hokkaido University, Kita 10, Nishi 8, Kita-ku, Sapporo 060-0808, Japan; 3Department of Magnetic and Biomedical Materials, Faculty of Materials Science and Technology, VNUHCM-University of Science, 227 Nguyen Van Cu Street, Ward 4, District 5, Ho Chi Minh City 70000, Vietnam; 4Ho Chi Minh City Campus, Vietnam National University, Linh Trung, Thu Duc, Ho Chi Minh City 70000, Vietnam

**Keywords:** cationic liposomes, charge ratio, guanine quadruplex, CpG oligodeoxynucleotides, cytokine induction

## Abstract

Cationic liposomes, specifically 1,2-dioleoyl-3-trimethylammonium-propane (DOTAP) liposomes, serve as successful carriers for guanine-quadruplex (G4) structure-based cytosine-guanine oligodeoxynucleotides (CpG ODNs). The combined benefits of CpG ODNs forming a G4 structure and a non-viral vector carrier endow the ensuing complex with promising adjuvant properties. Although G4-CpG ODN-DOTAP complexes show a higher immunostimulatory effect than naked G4-CpG ODNs, the effects of the complex composition, especially charge ratios, on the production of the pro-inflammatory cytokines interleukin (IL)-6 and interferon (IFN)-α remain unclear. Here, we examined whether charge ratios drive the bifurcation of cytokine inductions in human peripheral blood mononuclear cells. Linear CpG ODN-DOTAP liposome complexes formed micrometer-sized positively charged agglomerates; G4-CpG ODN-DOTAP liposome complexes with low charge ratios (0.5 and 1.5) formed ~250 nm-sized negatively charged complexes. Notably, low-charge-ratio (0.5 and 1.5) complexes induced significantly higher IL-6 and IFN-α levels simultaneously than high-charge-ratio (2 and 2.5) complexes. Moreover, confocal microscopy indicated a positive correlation between the cellular uptake of the complex and amount of cytokine induced. The observed effects of charge ratios on complex size, surface charge, and affinity for factors that modify cellular-uptake, intracellular-activity, and cytokine-production efficiency highlight the importance of a rational complex design for delivering and controlling G4-CpG ODN activity.

## 1. Introduction

CpG oligodeoxynucleotides (CpG ODNs) are short synthetic single-stranded DNA molecules containing unmethylated cytosine-guanine (CpG) motifs that trigger the immune system via toll-like receptor (TLR) 9 [1,2,3,4]. CpG ODNs directly activate TLR9-expressing cells, including human B cells and plasmacytoid dendritic cells (pDCs) [1,2,3,5], which, in turn, promote T helper1 cell responses and pro-inflammatory cytokines [6,7,8]. CpG ODNs also enhance antigen-specific humoral and cellular immune responses by improving the function of professional antigen-presenting cells [6,7,8]. These abilities of CpG ODNs endow them with great potential as vaccine adjuvants in the treatment of infectious diseases, cancers, and allergies.

However, linear CpG ODNs with a natural phosphodiester (PD) backbone are rapidly degraded by nucleases, which limits their clinical application [9]. While phosphorothioate-modified CpG ODNs display higher nuclease resistance, they still exhibit undesirable side effects such as prolonged time of coagulation, acute toxicity, and non-specific binding to proteins [10].

To overcome the limitations of using PD-backbone-based CpG ODNs as new drug modalities, we previously developed PD-backbone-based CpG ODNs with high nuclease resistance by using the guanine-quadruplex (G4) structure in the CpG ODN scaffold [11,12,13]. G4 is a four-stranded highly stable three-dimensional structure of ODNs that arises when guanine-rich sequences self-associate. This is characterized by the stacking of planar arrangements of four guanines, called G-quartets [14,15,16,17]. The insertion of CpG sequences into the loop region of G4 enhances the nuclease resistance of the CpG ODNs, resulting in high immunostimulatory activity of the G4-CpG ODNs in immune cells. Nevertheless, their uptake into cells is insufficient due to the electrostatic repulsion between the negatively charged phosphate groups on the ODN backbone and the cellular membranes.

To improve the cellular uptake of G4-CpG ODNs, we previously used a cationic liposome, the 1,2-dioleoyl-3-trimethylammonium-propane lipid (DOTAP) liposome, and cationic polymer, ε-poly-L-lysine (ε-PLL), as carriers for G4-CpG ODNs [18,19]. These complexes exhibited great immunostimulatory properties compared to naked G4-CpG ODNs both in vitro and in vivo. The cytokine profiles were also dependent on the composition or charge ratios of the G4-CpG ODN/carrier complexes. Interestingly, cytokine induction-switching has been observed using two different G4-CpG ODNs and different charge ratios of G4-CpG ODN-DOTAP complexes [18,19]. In particular, GD2 and GD3, which contain two or three CpG motifs in the second loop of G4 ODNs, showed different cytokine induction profiles. The GD2-DOTAP complex induced higher levels of interleukin (IL)-6 than the GD3-DOTAP complex. Conversely, GD2-DOTAP liposomes induced a significantly lower amount of interferon (IFN)-α than GD3-DOTAP liposomes in human peripheral blood mononuclear cells (hPBMCs). These results indicated that this switching did not depend on the number of CpG motifs but could rather be influenced by the loop length of the CpG ODNs. However, there is not enough information on how the difference in complex compositions or length of CpG ODNs affects the cytokine induction profiles.

In this study, we investigated the influence of the G4-CpG ODN-DOTAP liposome complex charge ratio on the cytokine profiles and the underlying mechanism in relation to intracellular behaviors. Complexes of G4-CpG ODNs and DOTAP liposomes were prepared at different charge ratios by varying the amount of DOTAP liposome in the complex with respect to the linear CpG ODNs and G4-CpG ODNs. We then examined the correlation of the charge ratio with complex size, zeta potential, cytokine induction, and cellular uptake.

## 2. Materials and Methods

### 2.1. Chemicals and Oligodeoxynucleotides

The sequences of all the oligodeoxynucleotides used in this study are given in Table 1. Linear CpG ODNs (CpG2, CpG3) were matched in both length and number of CpG motifs to the G4-CpG ODNs (GD2, GD3). To observe the cellular uptake of G4-CpG ODNs, we used GD3^5′Cy5^ and GD2^5′Cy5^, which were prepared by labeling the 5′-end of GD2 and GD3, respectively, with Cy5. All ODNs, including Cy5-labeled CpG ODNs with a PD backbone, were of high-performance liquid chromatography purity grade and were purchased from Eurofins Genomics (Tokyo, Japan). The cationic liposome solution, DOTAP liposomal transfection reagent (1 μg/μL), was obtained from Merck (Darmstadt, Germany).

### 2.2. G4 Structure Formation

The CpG ODNs were dissolved in sterile deionized water to prepare stock solutions with a concentration of ~100 µM that were stored at −30 °C until use. The G4 structures of the CpG ODNs were prepared using a previously reported method [11]. Briefly, ODNs were diluted from the stock solution using Dulbecco’s phosphate-buffered saline (D-PBS, DS Pharma Biomedical, Osaka, Japan). The ODN solutions then underwent a heat treatment process for G4 folding: incubation at 95 °C for 5 min, followed by slow cooling to 30 °C for 30 min, and then cooling to 4 °C using a thermal cycler (PCR Thermal Cycler Dice Standard TP650, Takara, Kyoto, Japan). The obtained ODN solutions with G4 structures were stored at 4 °C for further use.

### 2.3. Preparation and Characterization of the G4-CpG ODN-DOTAP Liposome Complexes

The charge ratio is a crucial factor for the properties of cationic liposome/DNA complexes. The charge ratio represents the molar ratio of positively charged amine groups in DOTAP to negatively charged phosphate groups in ODNs. The CpG ODN solutions with G4 structures were mixed with a constant volume of DOTAP liposomes to prepare G4-CpG ODN-DOTAP liposome complexes of various charge ratios. After 15 min of incubation at room temperature, the complex was immediately used for experiments. The final concentrations of G4-CpG ODNs and DOTAP liposomes in complexes of each charge ratio are summarized in Table 2.

The size distribution of the complexes (mean Z-average and polydispersity index) was measured using dynamic light scattering (DLS-8000; He-Ne laser; Otsuka Electronics, Osaka, Japan). The average particle size was calculated based on light intensity distributions using the CONTIN algorithm. The zeta potentials of the complexes were measured using a zeta potential analyzer (ELSZ-1000; Otsuka Electronics).

The topology of the G4-CpG ODNs, before and after complex formation with DOTAP liposomes, was analyzed based on circular dichroism (CD) spectra. Briefly, 2 μM CpG ODN solution in D-PBS was placed in a 1 cm path length quartz cuvette (model T-18-ES-10, Jasco, Tokyo, Japan). The spectra were recorded over a range of 230–320 nm at 25 °C with a scan rate of 50 nm/min, response time of 2.0 s, bandwidth of 1.0 nm, resolution of 0.2 nm, and sensitivity of 100 mdeg. The results shown for each sample represent the average of 10 scans. All spectra were smoothed using simple moving-average methods with 21 data points (±2 nm).

### 2.4. Quantification of G4-CpG ODN Loading onto DOTAP Liposomes

The G4 ODN loading efficiency of the complexes was indirectly assessed in terms of the concentration of unbound ODNs in the flowthrough following centrifugal (ultra) filtration of the complexes. First, we prepared GD2-DOTAP liposome complexes at an ODN concentration of 10 μM in D-PBS. An Amicon^®^ Ultra 100K device (Millipore, Bedford, MA, USA) was loaded with 200 µL of complexes, following which it was subjected to centrifugation for 20 min at 14,000× *g* and room temperature. The absorption spectra of the unbound ODNs in the flowthrough were recorded using a NanoDrop^®^ 2000 spectrophotometer (Thermo Fisher Scientific, Waltham, MA, USA). The concentration of the unbound ODNs was calculated from the standard curve of the free GD2 (0–20 μM) based on their absorbance values at the wavelength of 260 nm.

A gel electrophoretic mobility shift assay of different complexes was carried out using 10–20% linear gradient polyacrylamide gels (e-PAGEL gels, ATTO, Tokyo, Japan) that were run in tris-glycine buffer (25 mM Tris and 192 mM glycine) containing 4 mM potassium for 55 min at 21 mA and 4 °C.

### 2.5. Serum Stability Assay

The serum stability of the complexes was assessed to determine the resistance of the ODNs to nuclease degradation. First, free GD2 and GD2-DOTAP complexes with different charge ratios were mixed with 1× D-PBS containing fetal bovine serum (FBS) (Sigma-Aldrich, St. Louis, MO, USA). The final concentrations of GD2 and FBS in the assay were 0.5 μM and 50% (*v*/*v*), respectively. After incubation at 37 °C for 1, 2, 4, and 24 h, 3 μL ethylenediaminetetraacetic acid (0.25 M) was added to inactivate the DNase, followed by heating at 80 °C for 2 min. The treated complexes were run on 10–20% linear gradient polyacrylamide gels in tris-glycine buffer for 55 min at 21 mA and 4 °C. The ultra-low range DNA ladder 10 base pair (Thermo Fisher Scientific) was used as a marker. The bands in the gel were stained using SYBR^®^ Gold Nucleic Acid Gel Stain (Thermo Fisher Scientific) for 40 min. The fluorescence intensity of the non-degraded ODN band in the gel was quantified using Image Studio™ Lite 5.2 (Li-COR Biotechnology, Lincoln, NE, USA). The percentage of residual GD2 loaded into DOTAP liposomes was calculated using the following formula.
$$Residual\;ODNs\;\left(\%\right)=\frac{Intensity\;of\;Bands\;after\;24\;h\;}{Intensity\;of\;initial\;Bands\;\left(0\;h\right)}\times 100$$

### 2.6. Cell Culture

hPBMCs were purchased from Cellular Technology (Shaker Heights, OH, USA), prepared according to the manufacturer’s instructions, and resuspended in Roswell Park Memorial Institute (RPMI)-1640 medium (Thermo Fisher Scientific) containing GlutaMAX™, 4-(2-hydroxyethyl)-1-piperazinee-thanesulfonic acid (HEPES), 10% (*v*/*v*) heat-inactivated FBS, 100 U/mL penicillin, and 100 μg/mL streptomycin.

Human primary B cells and pDCs were isolated from hPBMCs using a B Cell Isolation Kit II (Miltenyi Biotec, Bergisch Gladbach, Germany) and an EasySep™ Human Plasmacytoid DC Isolation Kit (STEMCELL Technologies, Vancouver, BC, Canada), respectively, according to the manufacturer’s instructions.

PMDC05 cells, a human leukemic plasmacytoid dendritic cell line, were obtained from Niigata Technology Licensing Organization (Niigata, Japan) and cultured in Iscove’s Modified Dulbecco’s Medium (Thermo Fisher Scientific) containing 4 mM L-glutamine, 25 mM HEPES, 10% heat-inactivated FBS, 100 U/mL penicillin, and 100 μg/mL streptomycin.

Namalwa cells, a human B lymphocyte cell line (CRL-1432™), were obtained from American Type Culture Collection, USA, and maintained in RPMI-1640 medium (Fujifilm Wako Pure Chemical, Osaka, Japan) containing 7.5% heat-inactivated FBS, 100 U/mL penicillin, and 100 μg/mL streptomycin.

All cells were incubated in a humidified incubator maintained at 37 °C with 5% CO_2_.

### 2.7. Cell Viability Assay

Cell viability was determined based on the luminescence measured using a CellTiter Glo™ Luminescent Cell Viability Kit (Promega, Madison, WI, USA). This assay uses a luciferase reaction to measure ATP, which is present in metabolically active cells. Briefly, 1 × 10^6^ cells/well of hPBMCs were seeded into a 96-well plate and immediately stimulated with G4-CpG ODN-DOTAP complexes with different charge ratios at a final ODN concentration of 0.5 μM. D-PBS was used as a control. After incubation for 48 h, the cell supernatant was removed post-addition of CellTiter Glo Reagent. Luminescence signals were measured using a Plate Reader (Ensight, PerkinElmer, Waltham, MA, USA). The amount of luminescence determines the amount of ATP present, which is proportional to the number of viable cells.

### 2.8. Immunostimulatory Properties of the CpG ODN-DOTAP Liposome Complexes

hPBMCs were seeded at a density of 1 × 10^6^ cells/well (190 μL of medium) into a 96-well plate and treated with 10 µL of 10 µM free ODN solution or ODN-DOTAP liposome complexes with different charge ratios for 48 h. For the stimulation experiments, the final ODN concentration in the cell medium was 0.5 μM.

Primary B cells (1 × 10^5^ cells/well) and pDCs (5 × 10^3^ cells/well) were subjected to the same procedure of stimulation as hPBMCs.

PMDC05 and Namalwa cells were seeded into a 96-well plate (at densities of 4 × 10^5^ and 1 × 10^5^ cells/well, respectively) for 18 h. Afterward, 10 μL of free ODNs or ODN-DOTAP liposome complexes with different charge ratios were added to the medium for 6 h in the case of PMDC05 cells and 4 h in the case of Namalwa cells. For the stimulation experiments, the final ODN concentration in the cell medium was 1 μM.

After the indicated time, the supernatants were collected by means of centrifugation of the cells (at 10,000× *g*, 4 °C) for 10 min. To evaluate the cytokine expression levels in the supernatants, IL-6 Ready-Set-Go Kits (Thermo Fisher Scientific), an ELISA Flex Human IFN-α (HRP) Kit (MabTech, Stockholm, Sweden), and a Human IL-1β Uncoated ELISA Kit (Thermo Fisher Scientific) were used according to the manufacturer’s instructions.

To evaluate the relative transcript levels of the cytokines IL-6 and IFN-α in the cells, we performed reverse transcription/real-time quantitative polymerase chain reaction. Total RNA from the cells was extracted using ISOGEN (Nippon Gene, Tokyo, Japan) following the manufacturer’s instructions. The mRNA was purified using magnetic beads (RNAClean XP; Beckman Coulter, Brea, CA, USA). cDNA was synthesized from 0.5 μg of purified mRNA using ReverTra Ace™ qPCR RT Master Mix (Toyobo, Osaka, Japan) according to the manufacturer’s instructions. Quantitative real-time RT-PCR was performed using 50 ng of cDNA and a primer (FASMAC, Kanagawa, Japan), in a LightCycler 2.0 system (Roche, Switzerland), using GeneAce SYBR^®^ qPCR Mix α No ROX (Nippon Gene). The mRNA levels of the cytokine genes were normalized against those of glyceraldehyde 3-phosphate dehydrogenase as a housekeeping gene.

The primers used included: human GAPDH—forward, 5′-CATGGCCTCCAAGGAGTAAG-3′ and reverse, 5′-AGGGGTCTACATGGCAACTG-3′; human IL-6—forward, 5′-GCTGCAGGACATGACAACTC-3′ and reverse, 5′-TAAGTTCTGTGCCCAGTGGA-3′; and human IFN-α (consensus primer for IFN-α family)—forward, 5′-GTGAGGAAATACTTCCAAAGA-3′ and reverse, 5′-TCTCATGATTTCTGCTCTGACA-3′.

### 2.9. Confocal Laser Scanning Microscopy (CLSM)

CLSM was used to visualize the intracellular behavior of G4-CpG ODN-DOTAP liposome complexes in Namalwa and PMDC05 cells. A cell culture dish with a glass bottom (CELLview; Greiner Bio-One, Kremsmünster, Austria) was first coated with poly-L-lysine (Sigma-Aldrich, St. Louis, MO, USA). The cells were seeded into the dish at a density of 3.5 × 10^4^ cells/well, allowed to adhere for 18 h, and exposed to 5′-Cy5-labeled G4-CpG ODN-DOTAP liposome complexes for 2 h at 37 °C. Following that, the cells were washed three times with phosphate-buffered saline and stained using a MemBrite™ Fix Cell Surface Staining Kit (MemBrite™ Fix 488/515; Biotium, Fremont, CA, USA) according to the manufacturer’s protocol, followed by fixation with 4% paraformaldehyde for 10 min at room temperature. Before imaging, the cells were incubated with 4′,6-diamidino-2-phenylindole (DAPI) for 10 min (Thermo Fisher Scientific) for staining the nuclei. Cells incubated with free DNA served as a control. The cell images were obtained using a confocal laser scanning microscope (TCS SP5 II; Leica Microsystems, Wetzlar, Germany). Excitation wavelengths of 633, 488, and 360 nm were used for Cy5, MemBrite™, and DAPI, respectively. LAS AF (version 2.6.3. build 8173, Leica Microsystems, Wetzlar, Germany) was used to process the images.

### 2.10. Statistical Analysis

Differences between the groups were analyzed using one-way and two-way analysis of variance, followed by Tukey’s multiple comparisons tests. All statistical analyses were carried out using Prism version 8.2.0 for Windows (GraphPad Software, La Jolla, CA, USA).

## 3. Results

### 3.1. Preparation and Characterization of the G4-CpG ODN-DOTAP Liposome Complexes

To investigate the role of the charge ratio of CpG ODN-DOTAP liposome complexes in regulating the immune response, we prepared a series of CpG ODN-DOTAP liposome complexes with different charge ratios. Two G4-CpG ODNs, named GD2 and GD3, which contained two and three CpG motifs in the central loop region of the G4 structure, respectively, were used. The complex’s charge ratio was adjusted by changing the molar ratio of the cationic liposome-DOTAP liposome to the negatively charged single-stranded DNA-CpG ODN. The complexes of GD2 and GD3 with DOTAP liposomes are hereafter referred to as GD2-DOTAP and GD3-DOTAP, respectively. G4-CpG ODN loading, which relies on the electrostatic interaction between the positively charged amine head group on DOTAP and the negatively charged phosphate group on DNA, caused ODN condensation into the complexes.

First, we determined the loading efficiency of G4-CpG ODNs onto the DOTAP liposome. As shown in Figure 1a, at a charge ratio of 1, more than 50% of GD2 was loaded onto the liposome. In GD2-DOTAP with charge ratios of 1.5, 2, and 2.5, >80% of GD2 was complexed with DOTAP. The loading efficiency of GD2 onto the DOTAP liposome was also assessed using the gel electrophoretic mobility shift assay. As shown in Appendix A, with an increase in the charge ratio of the complex, the amount of GD2 in the sample decreased. This result indicated that the amount of G4-CpG ODNs loaded onto the surface of DOTAP increased with the charge ratio.

To identify the possible topological change in GD2 due to DOTAP binding, we investigated the CD spectra of naked GD2 and GD2-DOTAP complexes with different charge ratios (Figure 1b and Appendix A). The CD spectra of the complexes showed that G4 CpG ODN complexed with DOTAP partly maintained a hybrid-type G4 topology, characterized by positive peaks at wavelengths of 265 and 295 nm [20]. However, an elevation was observed in the peak height at a 265 nm wavelength as the charge ratios were increased, indicating that G4 CpG ODN formation involved a mixture of predominant anti-parallel loops or parallel loops. The CD spectrum of a random coil structure and linear CpG ODNs is characterized by a positive peak at 280 nm [12]. However, at charge ratios of 2 and 2.5, the CD spectra of the complexes resembled that of 1× D-PBS buffer with no discernible peaks, hinting that CpG ODN might not adopt a G4 structure, linear, or random coil and could thus primarily be in its aggregation form.

The size and zeta potential are crucial factors that influence cellular uptake and immune response [21]; hence, the hydrodynamic size and zeta potential of the G4-CpG ODN-DOTAP liposome complexes were determined at various charge ratios. Figure 1c and Appendix A show the hydrodynamic size of each complex as calculated using the CONTIN algorithm and the size distribution of complexes in terms of intensity-, volume-, and number-weighted particle sizes, respectively. The size and zeta potential of the DOTAP liposomes were approximately 130–135 nm and +40 mV, respectively. The sizes of the GD2-DOTAP were approximately 260, 333, 303, and 331 nm at charge ratios of 0.5, 1, 1.5, and 2, respectively. At a charge ratio of 2.5, there was a rapid increase in the hydrodynamic size. The complex tended to aggregate, with a diameter of 1500 nm at a charge of 2.5, and then started slowly decreasing at a higher charge ratio of 3 (Figure 1c). The high polydispersity index notably reflects these aggregations, indicating the presence of multiple aggregate populations. GD2-DOTAP liposome complexes remained stable and homogeneous at charge ratios of 0.5–1.5 for 1 week but showed significant aggregation after 2 weeks, as confirmed by PDI changes. In contrast, complexes at higher charge ratios agglomerated throughout the duration of the experiment (Appendix A). With an increasing charge ratio from 0.5 to 2, the zeta potential of the complexes was negative and gradually decreased (Figure 1d). This phenomenon was also observed in Wiethoff and Middaugh’s study [22]. The decrease in zeta potential is attributed to the high negative charge density on the complexes’ surfaces. The DNA becomes more densely packed on the surface of the complexes due to interactions with the cationic liposome. An increase in the zeta potential was observed at the higher charge ratios of 2.5 and 3, resulting from the extra addition of the cationic DOTAP liposome to GD2. The zeta potential of high-charge-ratio complexes reached close to a neutral charge. These charge-neutral complexes lack strong electrostatic repulsion between them, leading to rapid aggregation.

Similar results were observed in the case of GD3-DOTAP (Appendix A). The sizes of GD3-DOTAP were approximately 201, 196, and 203 nm at charge ratios of 0.5, 1, and 1.5, respectively. At a charge ratio of 2.0, there was a rapid increase in the hydrodynamic size. When the charge ratio of the GD3-DOTAP was increased to 2, the zeta potential of the GD3-DOTAP reached close to a neutral charge, which encouraged complex aggregation by decreasing the electrical repulsion between them. The GD3-DOTAP was positively charged at charge ratios above 2.5 and saturated at a value of +20 mV (Appendix A).

In contrast, in linear CpG ODNs with the same length and number of CpG motifs (linear CpG2 and linear CpG3, respectively), large aggregates of 0.7–3.8 µm were formed at all charge ratios (Appendix A). Therefore, we found that a G4 formation kept the size of the ODN-DOTAP complex uniformly small.

These results indicate that the charge ratio of the G4-CpG ODN-DOTAP liposome complex significantly affects its particle size and surface charge density. Furthermore, the G4 structure contributes to the formation of stable complexes that are negatively charged and <350 nm in size. This is a unique phenomenon that cannot be observed in the linear structures of CpG ODNs.

### 3.2. Nuclease Resistance of G4-CpG ODNs in Complex

DOTAP liposomes are known to protect the G4-CpG ODNs from DNase degradation [18]. Therefore, we examined the effect of the charge ratio on the nuclease resistance of GD2-DOTAP in high concentrations of serum known to contain nucleases. The relative amount of remaining ODNs was evaluated based on the intensity of the polyacrylamide gel electrophoresis bands after incubation for 1, 2, 4, and 24 h. The stability of the GD2-DOTAP was compared with that of naked GD2 in 50% (*v*/*v*) FBS at 37 °C. As shown in Figure 2, the naked GD2 was unstable and degraded approximately 85% in 50% (*v*/*v*) FBS after 24 h of incubation, whereas the GD2 in complexes with DOTAP liposomes remained more than the naked GD2 at all charge ratios. At charge ratios of 0.5 and 1, more than 50% of GD2 in the lipoplex was degraded after 24 h of incubation, whereas at charge ratios of 1.5, 2, and 2.5, over 85% of GD2 in the lipoplex still remained. Thus, an increase in the charge ratio resulted in increased nuclease stability of GD2 in the complexes.

### 3.3. Cytokine Induction by the G4-CpG ODN-DOTAP Complexes of Various Charge Ratios

In our previous study, DOTAP liposomes showed the potential to enhance the immunostimulatory activity of G4-CpG ODNs. Moreover, we found that the immunostimulatory property of the G4-CpG ODN-DOTAP liposome complexes (i.e., whether it predominantly induced IL-6 or IFN-α) was strongly dependent on the length of the loop containing the CpG motif in the G4 structure [18]. To examine the influence of spatial structure (random or G4 structure), length of CpG motif, and charge ratio of CpG ODN-DOTAP complexes on the induction of IL-6 and IFN-α, we designed variants of CpG ODNs with different structures and lengths. Two linear CpG ODNs, which matched the G4 CpG ODN length and number of CpG motifs (CpG2 and CpG3, respectively), were compared with two G4-CpG ODNs, GD2 and GD3. Four different types of CpG ODNs were loaded into cationic DOTAP liposomes of different charge ratios. The primary human immune cells, hPBMCs, were used to examine the immunostimulatory activity of the CpG ODN-DOTAP liposome complexes.

First, the cytotoxicity of the complexes was examined under the same conditions as those in the immunostimulatory assay. The relative viability of hPBMCs incubated with G4-CpG ODN-DOTAP liposome complexes of all charge ratios was approximately 100%, comparable with that of the control, D-PBS. This indicates that the complexes of G4-CpG ODNs and DOTAP liposomes exhibited no toxicity and maintained good biocompatibility (Appendix A).

The IL-6 and IFN-α production in hPBMCs exposed to CpG2-DOTAP, CpG3-DOTAP, GD2-DOTAP, or GD3-DOTAP complexes of different charge ratios is shown in Figure 3a,b. DOTAP liposomes alone do not show any cytokine induction [18]. The combination of CpG ODNs and DOTAP liposomes led to significant increases in cytokine induction in hPBMCs, thus confirming that the complex with DOTAP has the potential to enhance immunostimulatory activity. Cytokine levels produced by cells stimulated by G4-CpG ODN-DOTAP were significantly higher than those produced by cells stimulated by linear CpG ODNs. At low charge ratios of 0.5 and 1.5, the G4-CpG ODN-DOTAP liposome complexes markedly stimulated induction of IL-6 and IFN-α, reaching levels that were approximately 6 and 15 times greater, respectively, than those induced by linear CpG ODN-DOTAP complexes of identical charge ratios. Even though G4-CpG ODN-DOTAP liposome complexes of high charge ratios of 2 and 2.5 triggered modest cytokine induction, these levels still represented a 2–3-fold increase over the responses observed in the case of linear CpG ODN-DOTAP liposome complexes. These results are consistent with previous reports that G4 formation is beneficial for enhancing the immunostimulatory activity of the CpG ODN-DOTAP complex [18,19].

GD3-DOTAP induced higher or comparable levels of IL-6 and IFN-α than GD2-DOTAP. This can be explained by the central loop length of GD2 and GD3 (i.e., 14 and 22 mer, respectively), since we have previously reported that, although the number of CpG motifs does not affect the cytokine induction of G4-CpG ODNs, there is an influence of the loop length on the same [18].

Furthermore, we investigated the effect of the charge ratio on the immunostimulatory activity. GD2-DOTAP and GD3-DOTAP complexes of different charge ratios showed similar tendencies of cytokine induction. We found a correlation between the charge ratios and cytokine inductions mediated by the G4-CpG ODN-DOTAP liposome complexes. Cells treated with G4-CpG ODN-DOTAP liposome complexes of low charge ratios (0.5 and 1.5) induced significant production of IL-6 and IFN-α simultaneously, while both IL-6 and IFN-α secretion levels were reduced upon treatment with G4-CpG ODN-DOTAP liposome complexes of high charge ratios (2 and 2.5). These findings indicate that cytokine induction levels can be easily altered by changing the charge ratio.

We also incorporated a more detailed time-course analysis of cytokine production following GD2-DOTAP administration at a charge ratio of 1.5 (Appendix A). The observed trends indicate that both IL-6 and IFN-α levels rise progressively over time. During the initial stages (at 2 and 6 h), the GD2-DOTAP complex did cause a moderate increase in IL-6; however, this complex induced minimal to no IFN-α during this period. This elevation became even more apparent after 12 h and sustained its peak up to 72 h. These findings are illustrative of the trafficking of the complex in cells. After cellular uptake, there appears to be a latency in the release of ODN and its subsequent translocation to the endosome/lysosome, where it encounters the active TLR9, leading to robust cytokine production.

The G4-CpG ODN-DOTAP liposome complex charge ratio-mediated regulation of bifurcated cytokine inductions of IL-6 and IFN-α was observed in hPBMCs in this study, which contain various types of cells with different densities. As B cells and pDCs have been identified in the human blood as expressing TLR9 and thus responding directly to CpG ODNs [5], we isolated these cells from PBMCs to investigate the effect of CpG ODNs on them. As shown in Figure 3c, GD2-DOTAP with low charge ratios of 0.5 and 1.5 induced significantly high levels of IL-6 in B cells, while completely failing to promote IL-6 at higher charge ratios of 2 and 2.5. A similar tendency was observed for IFN-α induction in pDCs by complexes of different charge ratios (Figure 3d). These results suggested that the influence of charge ratios on IL-6 and IFN-α production in hPBMCs mainly occurred in B cells and pDCs, respectively. This correlation is strongly attributed to the size of the complex [23,24].

### 3.4. Cellular Uptake of G4-CpG ODN-DOTAP Complexes of Different Charge Ratios

The efficient immunostimulatory activity of CpG ODNs depends on their cellular uptake, which is largely influenced by their particle size and the surface charge of the liposomes and DNA/liposome complexes. In Section 3.1, we found that the charge ratio influenced the size and surface charge of the G4-CpG ODN-DOTAP liposome complexes. In this part of the study, we examined the internalization of complexes of different charge ratios into immune cell lines. First, we examined the behavior of the G4-CpG ODN-DOTAP complexes using immune cell lines to determine whether these induced a similar tendency of cytokine production in hPBMCs. We used Namalwa (a human B lymphocyte cell line) and PMDC05 (a human leukemic pDC line) as models of primary B cells and pDCs, respectively. These cells were stimulated with G4-CpG ODN-DOTAP complexes of different charge ratios. As shown in Appendix A, the relative mRNA levels of IL-6 and IFN-α induced by GD3-DOTAPs were similar to those in hPBMCs. GD3-DOTAP induced significantly higher IL-6 and IFN-α mRNA expression in the cells as compared to that induced by naked GD3. We also observed that the transcript levels of IL-6 and IFN-α were associated with the charge ratio of the G4-CpG ODN-DOTAP liposome complex in these cell lines. The complexes with the low charge ratios induced significant levels of IL-6 and IFN-α, which started to decrease as the charge ratios increased.

Confocal microscopy was used to visualize the intracellular localization of the G4-CpG ODN-DOTAP liposome complexes. G4-CpG ODNs were labeled with red fluorescein-Cy5 and then complexed with DOTAP liposomes at different charge ratios. A similar tendency was observed in both the Namalwa and PMDC05 cell lines (Figure 4 and Figure 5, respectively). When naked GD2^5‘Cy5^ was employed, after 2 h, there were no signs of naked GD2^5‘Cy5^ in these cells, indicating low cellular uptake. However, GD2-DOTAP at charge ratios of 0.5 and 1.5 was localized inside the cells and exhibited stronger red fluorescence, indicating much better cellular uptake efficiency. Contrary to the high uptake of low-charge-ratio complexes, there was weak fluorescence or no signs inside the cells incubated with high-charge-ratio complexes. The complexes with high charge ratios of 2 and 2.5 were bound on the cell membrane, indicating a low cellular uptake. GD3-DOTAP of different charge ratios showed similar cellular uptake efficiency in Namalwa and PMDC05 cells (Appendix A). The complexes with low charge ratios showed high cellular uptake and were located inside the cells, while the complexes with high charge ratios of 2 and 2.5 caused aggregation of the cells, resulting in the observation of saturated red fluorescence. This indicated that the cellular uptake efficiency was influenced by the charge ratio of the G4-CpG ODN-DOTAP liposome complex. The difference in the particle size and surface charge of the complexes with different charge ratios possibly correlated with the binding and uptake of the complexes into the cells.

### 3.5. Cytokine Induction by GpC ODN and Methylated CpG ODN-DOTAP Liposome Complexes in Human PBMCs

The IL-6 production observed in the hPBMCs upon treatment with G4-CpG ODN-DOTAP liposome complexes of high charge ratios (2 and 2.5) (Figure 3a) did not correspond to the IL-6 production in the B cells (Figure 3c). Thus, it appears that the IL-6 generated by G4-CpG ODN-DOTAP liposome complexes with high charge ratios in hPBMCs does not originate from B cells or pDCs, which mainly express TLR9 in human cells.

To further elucidate the immunostimulatory effect of the G4-CpG ODN-DOTAP liposome complexes of different charge ratios, we examined the activities of the complexes with methylated GD2 and GpC GD2, which are not recognized by TLR9 [18]. We assessed the immunostimulatory activity of GD2-DOTAP, methylated GD2-DOTAP, and GpC GD2 -DOTAP of different charge ratios in hPBMCs. As shown in Figure 6, the cytokine production induced by the G4-CpG ODN-DOTAP complexes relied on the presence of CpG motifs, as there was cessation of cytokine induction when these CG sequences were substituted with GC. Interestingly, at higher charge ratios of 2 and 2.5, the complexes with methylated GD2 were still capable of inducing IL-6 at levels comparable to those induced by GD2. This result suggests that, when delivered by DOTAP liposomes of a high charge ratio, G4-CpG ODNs stimulate IL-6, regardless of the methylation status.

Meanwhile, IFN-α secretion is strongly dependent on TLR9. GpC GD2-DOTAP and methylated GD2-DOTAP eliminated the induction of IFN-α at all charge ratios (Figure 6b). This result suggests that the induction of IFN-α by the ODN-DOTAP liposome complex is entirely dependent on the unmethylated CpG sequence.

## 4. Discussion

Extensive research has been carried out to identify the parameters that influence the efficiency of liposomes as a delivery system for DNA. It has been determined that the physicochemical properties of DNA/liposome complexes play a significant role in the effectiveness of liposomes. Various factors, such as particle size, surface charge, liposome composition, and DNA/liposome charge ratio, have been examined for their impact on transfection efficiency, gene expression, and immunostimulatory activity [25,26,27,28,29]. Among these factors, the charge ratio of the DNA/liposome complex is a critical element. In the context of employing liposomes as a vaccine adjuvant carrier to enhance vaccine efficacy, there is no direct evidence illustrating the effect of lipoplex charge ratios on cytokine induction. In this study, we aimed to investigate the influence of the charge ratio on cytokine induction by the G4-CpG ODN-DOTAP liposome complex and further elucidate the underlying mechanisms that contribute to the effectiveness of liposome-based delivery systems.

The types of cytokines induced by CpG ODNs depend on their structural properties [30]. Class A CpG ODNs (CpG-A), which contain a palindrome sequence and form a nanometer-sized high-order structure, mainly induce the secretion of type I IFNs from pDCs [29,30,31], while Class B CpG ODNs (CpG-B), which form random structures, stimulate B cells to produce pro-inflammatory cytokines such as tumor necrosis factor-α, IL-6, and IL-12 [30,32]. Interestingly, when complexed with a DOTAP liposome, CpG-A stimulated the production of type I IFNs in B cells and conventional dendritic cells, which are not typically known to produce type I IFNs upon stimulation by CpG-A [29,33]. The CpG-B-DOTAP complex has also been shown to enable the activation of endosomal TLR9 and induce type I IFN production in pDCs [29]. IL-6 and IFN-α play important roles in the immune response. IL-6 induces B cell maturation, leading to enhanced production of antibodies such as IgM and IgG2a [34], while IFN-α enhances the cytotoxic activity of natural killer cells and certain T cells, promotes the differentiation of immature dendritic cells, and boosts the adaptive immune response [35]. Therefore, the bifurcation of these cytokine inductions is necessary for the effectiveness of immunotherapy. It has been reported that the bifurcation of these cytokine inductions by CpG ODNs is affected by several factors, including the binding mode of CpG ODNs onto nanoparticles [36], surface charge of nanoparticles [37], and sequence of CpG ODNs [18].

Before exploring cellular events, we characterized the complexes. Different charge ratios of complexes significantly affect the particle size of the liposomes, since the nucleic acid chain length does not affect complex sizes [38]. Linear CpG ODN-DOTAP liposome complexes formed a large aggregate at all charge ratios. The linear structure allows for multiple binding sites along the length of the ODN, facilitating the creation of large aggregates as multiple molecules bind together. In addition, the resulting aggregates might also be stabilized by additional interactions between DOTAP molecules, leading to a larger complex. On the other hand, G4-CpG ODN-DOTAP liposome complexes with low charge ratios between 0.5 to 1.5 had a relatively small particle size (<350 nm) and only formed microparticles (>1200–2200 nm) at higher charge ratios. The zeta potential of G4-CpG ODN-DOTAP liposome complexes first decreased due to the high negative charge density on the surface, following which it increased upon addition of an increasing amount of the cationic liposome. At this point, the net charge on the surface of the complexes becomes neutral, resulting in no electrical barrier to aggregation between complexes. The complexes tend to adhere to each other due to Van der Waals attraction, resulting in larger-sized complexes [39,40,41,42].

Endocytosis is the main pathway for a DNA/liposome complex to enter the cells. The particle size and surface charge of the liposomes and DNA/liposome complexes largely influence their intracellular behaviors, which have been shown to affect immunostimulatory activity [26,28,29,43]. After confirming that the regulation of bifurcated cytokine inductions was mainly induced in human B cells and pDCs (Figure 3c,d), we used Namalwa and PMDC05 cells as models to investigate the internalization of G4-CpG ODN-DOTAP liposome complexes. The complexes at low charge ratios (0.5 and 1.5), with their small size and negative surface charge (<350 nm, −40 mV to −10 mV), showed higher cellular uptake and were located inside cells, resulting in cytokine induction (Figure 4 and Figure 5). This finding is consistent with a previous study that reported that 100–500 nm-sized particles induce both IL-6 and IFN-α production [26]. Meanwhile, as the charge ratio increased, the complexes became larger (>1000 nm) and were hardly taken into the cells, leading to reduced cytokine secretion. This result suggested that the difference in cellular association of low and high charge ratios was size-dependent. However, the high-charge-ratio complexes with positive surface charge (>20 mV) were still observed on the cell membrane due to electrostatic attraction, indicating surface-charge dependency. Therefore, the regulation of cytokine induction could be explained by the cellular uptake efficiency of G4-CpG ODN-DOTAP liposome complexes of different charge ratios.

Because the micrometer-sized complexes were not taken up by dendritic cells and B cells, it was assumed that the CpG ODN-DOTAP liposome complex with a high charge ratio did not induce IL-6 and IFN-α production. However, both GD2-DOTAP and GD3-DOTAP complexed with a charge ratio of 2.5 still induced significant levels of both IL-6 and IFN-α, as compared to naked ODNs from hPBMCs (Figure 3a,b). Furthermore, the GD2-DOTAP with a charge ratio of 2.5 did not induce both IL-6 and IFN-α from isolated B cells and pDC cells, respectively. We also confirmed that the ability of the G4-CpG ODN-DOTAP complex to stimulate an immune response was highly reliant on the presence of CpG motifs (Figure 6). Chen et al. reported that, upon binding of CpG ODN 2216 (a Class A CpG ODN known to induce a high amount of IFN-α in the mouse dendritic cell line BC-1) to 3 μm particles, the resulting complex completely failed to promote IFN-α production [26]. The positive surface charge of G4-CpG ODN-DOTAP liposome complexes with a high charge ratio could effectively promote the maturation of dendritic cells [28]. Moreover, this discrepancy may arise from the inherent advantages of the G4 structure and non-viral vector carrier DOTAP [18]. The capacity to induce IFN-α production has been recognized to correlate with the potential to form extensive ODN structures [44].Moreover, the high-charge-ratio complexes with large sizes are taken into macrophages via micropinocytosis [45]. Besides B cells and pDCs, the high-charge-ratio complexes might take up natural killer cells, monocytes/macrophages, and T cells in hPBMCs as well, leading to secreted IL-6 production. Yasuda et al. reported that macrophages were activated by the DNA/cationic liposome complex via TLR9-dependent and -independent pathways [46]. There are two TLR9-independent pathways that secrete pro-inflammatory cytokines, including cyclic GMP-AMP synthase-stimulator of interferon genes (cGAS-STING) and src-family kinase-mediated signaling pathways [47]. In TLR9-negative cells, Class A CpG ODNs have been shown to significantly induce IFN-α and a low level of IL-6 by activating the cGAS-STING signaling pathway [48], while Sanjuan et al. observed that CpG ODN activates src-family kinases at the plasma membrane following upstream MyD88 activation, thereby promoting IL-6 induction [47]. We observed that IL-1β secretion was induced by the high-charge-ratio complexes (Appendix A). IL-1β, a critical mediator of the inflammatory response, is processed via the activation of the Nod-like receptor family pyrin domain-containing protein 3 inflammasome [49]. IL-1β triggers the induction of pro-inflammatory cytokines, including IL-6 [50]. Therefore, the production of IL-6 may be a consequence of the local inflammation caused by the high-charge-ratio complexes. However, our research has not clarified the mechanism by which the large micrometer-sized CpG ODN-DOTAP liposome complex induces IL-6 and IFN-α. Thus, there is a need for future studies to identify the specific cells in PBMCs that secrete IL-6 and IFN-α cytokines, or to verify if the cytokines are indeed induced synergistically by multiple cells in PBMCs.

## 5. Conclusions

In conclusion, we clarified that the regulation of cytokine induction by the G4-CpG ODN-DOTAP liposome complex in immune cells is strongly dependent on the charge ratio of the complex. A change in the charge ratio was found to affect the size, surface charge, and topology of the G4-CpG ODN-DOTAP liposome complex, leading to direct alterations in the cellular uptake, intracellular behaviors, and efficiency of cytokine induction. Thus, this study provides information for the rational design of a G4-CpG ODN-DOTAP liposome complex that allows for controlling cytokine profiles.

## Figures and Tables

**Figure 1 biomolecules-13-01639-f001:**
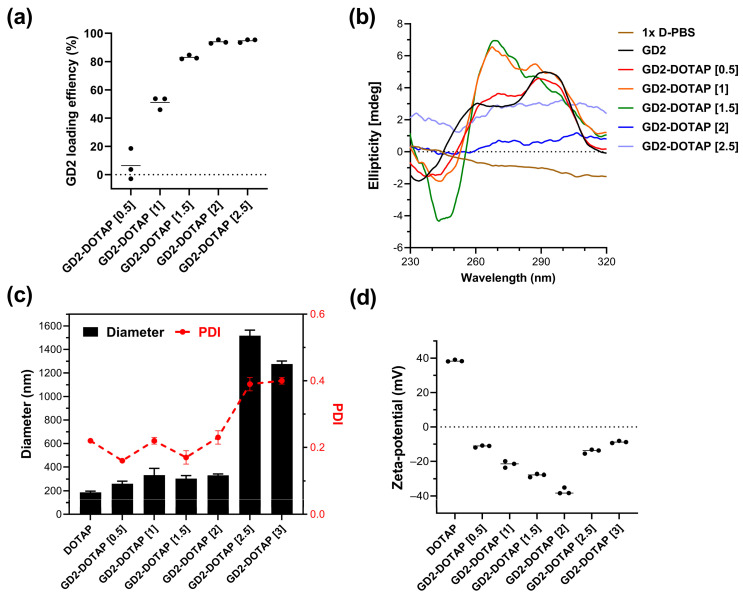
Characterization of the G4-CpG ODN-DOTAP liposome complexes with various charge ratios. (**a**) GD2 loading efficiency onto DOTAP liposome. (**b**) Circular dichroism spectra of naked GD2 and GD2-DOTAP liposome complexes. (**c**) Hydrodynamic size (black bar), polydispersity index (red symbol), and (**d**) zeta potential of the GD2-DOTAP liposome complexes. Data are presented as mean ± SD (*n* = 3).

**Figure 2 biomolecules-13-01639-f002:**
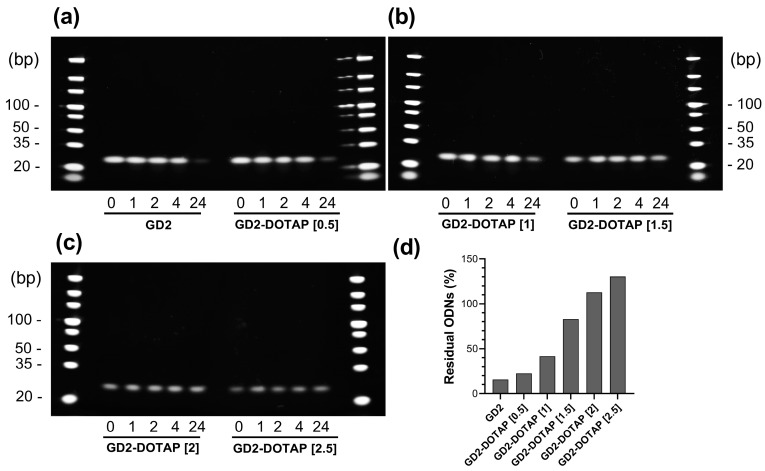
Nuclease resistance of naked GD2 and GD2-DOTAP complexes of various charge ratios in 50% fetal bovine serum after incubation for 1, 2, 4, and 24 h. (**a**–**c**) Images of polyacrylamide gel electrophoresis. (**d**) Quantitative results of the G4-CpG ODNs remaining after incubation in 50% serum for 24 h.

**Figure 3 biomolecules-13-01639-f003:**
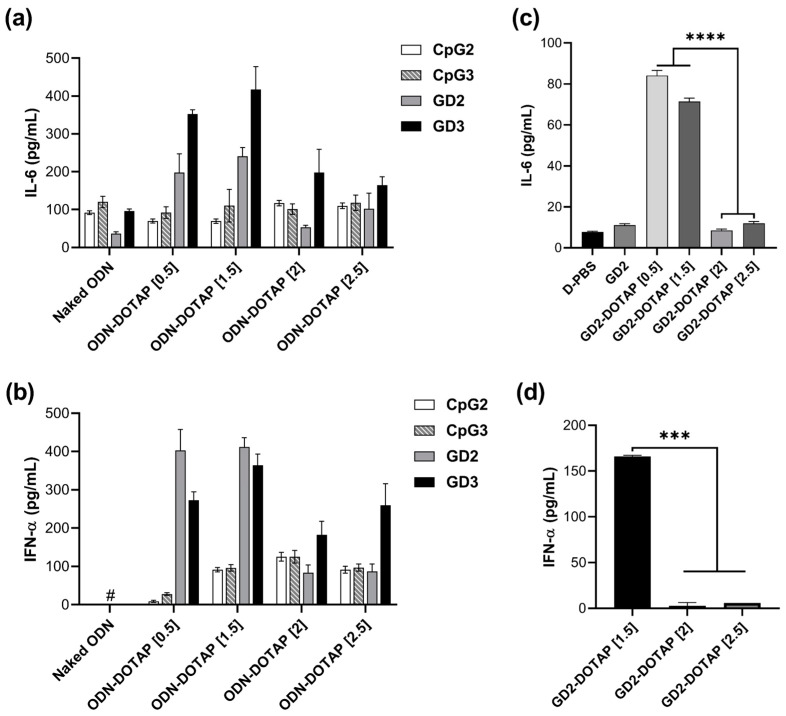
Effects of G4 structure and charge ratio of the CpG ODN-DOTAP liposome complexes on the production of IL-6 and IFN-α. (**a**,**b**) PBMCs, (**c**) B cells, and (**d**) pDCs were stimulated with 0.5 µM naked CpG ODNs or CpG ODN-DOTAP liposome complexes for 48 h. Data are represented as mean ± SD (*n* = 3–5). **** *p* < 0.0001 and *** *p* < 0.001 (one-way analysis of variance, followed by Tukey’s multiple comparisons test). # lower than the detection limit (3.9 pg/mL).

**Figure 4 biomolecules-13-01639-f004:**
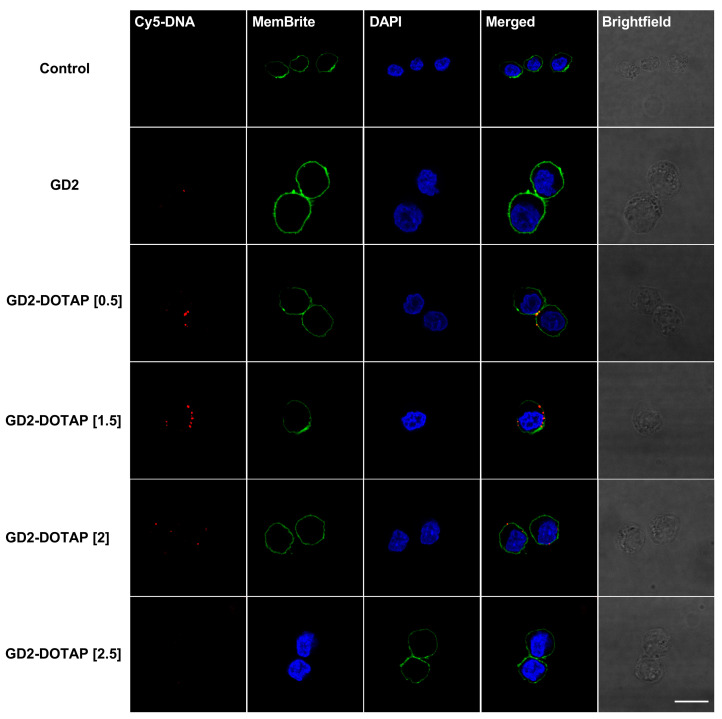
Internalization of naked GD2 and GD2-DOTAP complexes into Namalwa cells. The cells were incubated with 0.5 μM GD2^5′Cy5^ or GD2^5′Cy5^-DOTAP for 2 h. The complexes with low charge ratios of 0.5 and 1.5 underwent higher cellular uptake and were located inside the cells, as compared to naked GD2 or complexes with high charge ratios. Non-treated Namalwa cells were used as the negative control. GD2 was labeled with Cy5 (red). The cell membranes and nuclei were stained with MemBrite™ (green) and DAPI (blue), respectively. Scale bar: 10 μm.

**Figure 5 biomolecules-13-01639-f005:**
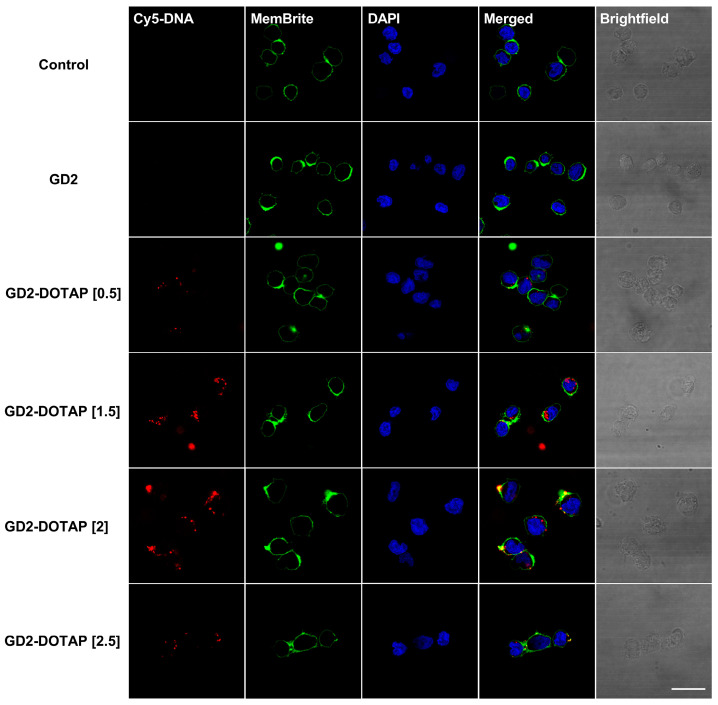
Internalization of naked GD2 and GD2-DOTAP liposome complexes into PMDC05 cells after stimulation for 2 h. GD2-DOTAP liposome complexes of low charge ratios (0.5 and 1.5) were located inside the cells, while GD2-DOTAP of high charge ratios (2 and 2.5) caused aggregation of the cells. Non-treated cells were used as the control. Cy5 (red), MemBrite™ (green), and DAPI (blue) represent GD2, the cell membrane, and nuclei, respectively. Scale bar: 10 μm.

**Figure 6 biomolecules-13-01639-f006:**
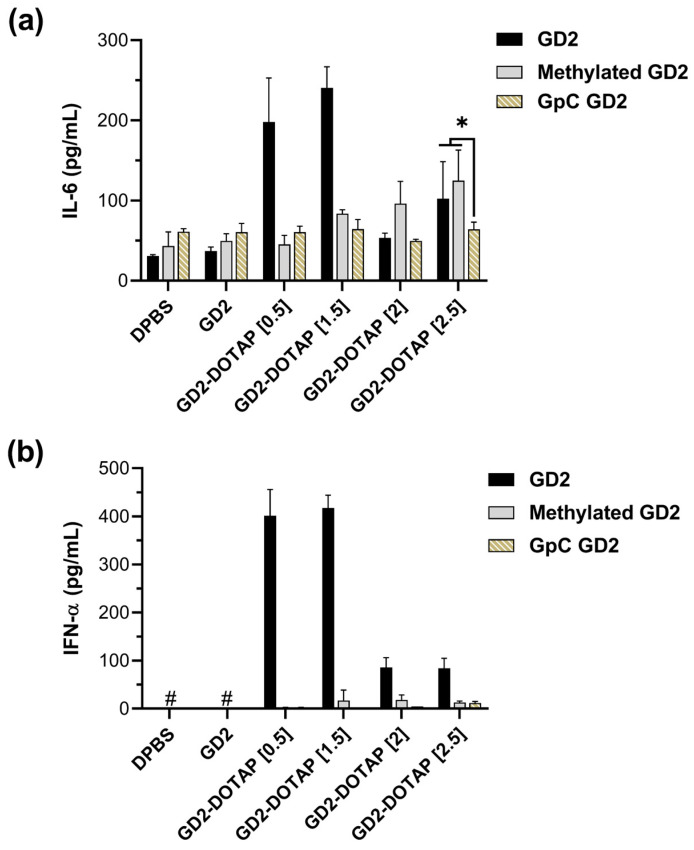
Cytokine induction by DOTAP liposome complexed with GD2, cytosine-methylated GD2 (methylated GD2), and GD2 with CG sequences replaced by GC sequences (GpC GD2) in hPBMCs after 48 h stimulation. (**a**) IL-6 production, (**b**) IFN-α production. Data are represented as mean ± SD (*n* = 5). * *p* < 0.05 (one-way analysis of variance, followed by Tukey’s multiple comparisons test); # = lower than the detection limit (3.9 pg/mL).

**Table 1 biomolecules-13-01639-t001:** Sequences of oligodeoxynucleotides used in this study.

Name	Sequence from 5′ to 3′	Length (mer)	Structure
GD2	**GGG**TT**GGG**GTCGTTTTGTCGTT**GGG**TT**GGG**	30	G4
GD3	**GGG**TT**GGG**GTCGTTTTGTCGTTTTGTCGTT**GGG**TT**GGG**	38	G4
CpG2	GTGTTGTGGTCGTTTTGTCGTTGTGTTGTG	30	Linear
CpG3	GTGTTGTGGTCGTTTTGTCGTTTTGTCGTTGTGTTGTG	38	Linear
Methylated GD2	**GGG**TT**GGG**GTC_Met_GTTTTGTC_Met_GTT**GGG**TT**GGG**	30	G4
GpC GD2	**GGG**TT**GGG**GTGCTTTTGTGCTT**GGG**TT**GGG**	30	G4
ssODN	GTCGTTTTGTCGTTTTGTCGTTTTGTCGTT	30	Linear

Bold letters, G-tract; underline, unmethylated CpG motif; C_Met_, cytosine methylation.

**Table 2 biomolecules-13-01639-t002:** Formulation of ODN-DOTAP liposome complexes with different charge ratios.

Charge Ratio	0.5	1	1.5	2	2.5	3
The concentration of ODNs in the complex solution (μM)	10					
The concentration of DOTAP in the complex solution (μM)	0.15	0.3	0.45	0.6	0.75	0.9

## Data Availability

Data are contained within the article.

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
