# Peer review of "Influence of the Charge Ratio of Guanine-Quadruplex Structure-Based CpG Oligodeoxynucleotides and Cationic DOTAP Liposomes on Cytokine Induction Profiles"

_biomolecules, 2023, doi:10.3390/biom13111639_

Round 1
Reviewer 1 Report
Comments and Suggestions for Authors
The manuscript by Le at al advances the understanding of CpG motif containing G4 structures, complexed for delivery with DOTAP, as immune stimulants. Overall the work demonstrated is well presented and mostly support the stated conclusions, but there are notable areas of concern related to improved controls, wider understanding of cytokine profiles, and a lack of support for a major conclusion related to TLR9. Addressing these concerns will significantly improve the manuscript both in soundness and in impact on a larger scientific community.
Major concerns:
The comparison of DNA as “linear” versus “G4” is misleading given the oligonucleotides employed in the study. In particular, CpGx2 and CpGx3, the linear ODN’s, represent only the central loop constituents. They are not matched in length or complexity to the longer G4 forming sequences. Proper linear controls for this study need to include G-to-T (or A) substitutions within the G-triplets of GD2 and of GD3.
The data presented make a strong case for GD2-DOTAP at a 1.5 charge ratio being the most promising immune stimulant candidate. The manuscript would be significantly enhanced by a more broad evaluation of cytokine changes (both pro- and anti-inflammatory) with this species at a particular time point. The most likely candidate for the time point is 48 h, given the effects seen in Figure 3, but a time course for IL-6 and IFN-g with this species in PBMCs could better inform the time.
On the note of time - there is a wide variation in exposure time within the presented experiments that make comparisons not feasible. Figure 3 shows 48 h data, whereas Fig 4 and 5 are after 2 or 6 h of exposure and the time of exposure in figure 6 is unclear. These need to either be justified or amended.
The comparison of secreted cytokine levels from PBMCs and mRNA changes in the other two cell lines is not ideal. The authors either need to justify the switch to mRNA or compare secreted profiles.
Lastly, regarding Figure 6, the data presented do not support the conclusions of a TLR-9 independent mechanism. Methylation of the DNA does eradicate both IL-6 and IFN-g secreted profile changes associated with GD2-DOTAP.
Minor concerns:
The description of G4 formation as mixed or parallel isn’t accurate based on the ECD presented. All structures noted are mixed, they are just mixed with anti parallel loops or parallel loops predominating.
The DNA integrity studies utilized a gel dye, but if the DNA is Cy5 labeled - why?
Figure 1D is presumably comparing the effects of DNA integrity after 24 h, but that isn’t clear. Please clarify. Also, please justify this time point by quantifying the DNA remaining with the uncomplexed GD2 first.
Reviewer 2 Report
Comments and Suggestions for Authors
See attached review.

Reviewer 3 Report
Comments and Suggestions for Authors
The authors presente a thorough characterization of lipoplexes formed by DOTAP liposome and G4-ODN in which the loop have been modified to insert 2 or 3 CpG sequences and on their effects on cytokine inductions as a function of the charge ratio between liposome and DNA. Overall the paper is clear and results convincing, some minor comments on the use of terms DOTAP and liposome, as the complexes are composed by ODN and preformed DOTAP liposomes the use of term DOTAP (in the title too) can be misleading.
Quality of CD spectra is quite poor particularly considering they are the average of 10 scans.
In the presence of multiple aggregate populations and high PDI values the zeta potential value can have little meaning, Authors should ate least warn the readers or comment on this.
Round 2
Reviewer 2 Report
Comments and Suggestions for Authors
The authors essentially addressed all the major issues raised by the Reviewers. I have no additional concern and in my opinion this article can be accepted for publication.